# Protective Role of Recombinant Human Thrombomodulin in Diabetes Mellitus

**DOI:** 10.3390/cells10092237

**Published:** 2021-08-29

**Authors:** Yuko Okano, Atsuro Takeshita, Taro Yasuma, Masaaki Toda, Kota Nishihama, Valeria Fridman D’Alessandro, Chisa Inoue, Corina N. D’Alessandro-Gabazza, Tetsu Kobayashi, Yutaka Yano, Esteban C. Gabazza

**Affiliations:** 1Department of Immunology, Faculty and Graduate School of Medicine, Mie University, Tsu 514-8507, Mie, Japan; yu.higaco@gmail.com (Y.O.); johnpaul0114@yahoo.co.jp (A.T.); t-yasuma0630@clin.medic.mie-u.ac.jp (T.Y.); t-masa@doc.medic.mie-u.ac.jp (M.T.); immunol@doc.medic.mie-u.ac.jp (V.F.D.); dalessac@clin.medic.mie-u.ac.jp (C.N.D.-G.); 2Department of Diabetes and Endocrinology, Faculty and Graduate School of Medicine, Mie University, Tsu 514-8507, Mie, Japan; kn2480@gmail.com (K.N.); chisa.i.0417@gmail.com (C.I.); yanoyuta@clin.medic.mie-u.ac.jp (Y.Y.); 3Department of Pulmonary and Critical Care Medicine, Faculty and Graduate School of Medicine, Mie University, Tsu 514-8507, Mie, Japan; kobayashitetsu@hotmail.com

**Keywords:** insulin resistance, diabetes mellitus, apoptosis, thrombomodulin, glucose intolerance, immune cells

## Abstract

Diabetes mellitus is a global threat to human health. The ultimate cause of diabetes mellitus is insufficient insulin production and secretion associated with reduced pancreatic β-cell mass. Apoptosis is an important and well-recognized mechanism of the progressive loss of functional β-cells. However, there are currently no available antiapoptotic drugs for diabetes mellitus. This study evaluated whether recombinant human thrombomodulin can inhibit β-cell apoptosis and improve glucose intolerance in a diabetes mouse model. A streptozotocin-induced diabetes mouse model was prepared and treated with thrombomodulin or saline three times per week for eight weeks. The glucose tolerance and apoptosis of β-cells were evaluated. Diabetic mice treated with recombinant human thrombomodulin showed significantly improved glucose tolerance, increased insulin secretion, decreased pancreatic islet areas of apoptotic β-cells, and enhanced proportion of regulatory T cells and tolerogenic dendritic cells in the spleen compared to counterpart diseased mice treated with saline. Non-diabetic mice showed no changes. This study shows that recombinant human thrombomodulin, a drug currently used to treat patients with coagulopathy in Japan, ameliorates glucose intolerance by protecting pancreatic islet β-cells from apoptosis and modulating the immune response in diabetic mice. This observation points to recombinant human thrombomodulin as a promising antiapoptotic drug for diabetes mellitus.

## 1. Introduction

The growing number of patients with diabetes mellitus (DM) is a serious human health concern worldwide [1]. The global diabetic population is estimated to be about 460 million [2]. The most frequent causes of DM-associated death are complications of small (retinopathy, nephropathy, neuropathy) and large (stroke, coronary heart disease, lower extremity arterial disease) vessels [3,4,5]. Type 1 DM causes selective destruction of the pancreatic islet β-cells by genetic- and autoimmune-mediated mechanisms, leading to severe insulin deficiency [6,7]. Type 2 DM is associated with lifestyle and genetic factors that cause insulin resistance, impaired biosynthesis and secretion of insulin, and reduced β-cell mass, resulting in a relative insufficiency of insulin activity [8]. In both types of DM, a persistent hyperglycemic condition leads to excessive oxidative stress, endoplasmic reticulum stress, and autophagy dysregulation that ultimately result in the apoptosis of β-cells [9,10]. The death of β-cells triggers a vicious cycle of reduced insulin secretion, hyperglycemia, and increased oxidative stress that results in vasculopathy [9,10]. Therefore, β-cell apoptosis is considered one of the major mechanisms in DM pathogenesis. The current treatment of DM is only symptomatic [11,12]. Many drugs that can control hyperglycemia are currently available for treating diabetic patients [11,12]. However, to date, no drug that can suppress β-cell apoptosis has been developed.

Thrombomodulin (TM) is a cell-membrane-bound glycoprotein expressed by multiple cells, including vascular endothelial cells [13]. Structurally, TM contains three extracellular domains, one transmembrane domain, and one cytoplasmic tail domain [13,14]. Thrombin, a procoagulant factor that cleaves fibrinogen to fibrin during coagulation activation, binds to the epidermal growth factor-like domain of TM [13,15]. After binding to TM, thrombin loses its procoagulant property to promote anticoagulant activity by converting the anticoagulant protein C to its active form activated protein C (APC) [13]. Apart from its anticoagulant activity, TM also inhibits inflammation and the immune response by blocking the high-mobility group protein B-1 (HMGB1), by suppressing the activity of immune cells and the activation of the complement system, and by inhibiting cell apoptosis via its G-protein-coupled receptor 15 [13,14,16,17]. TM also promotes antioxidant activity by enhancing the nuclear factor (erythroid-derived 2)-like 2 (NRF2) nuclear translocation antioxidant pathway [18]. Several in vivo animal experimental studies and clinical trials have also recapitulated TM’s beneficial properties. TM has preventive effects in diabetic renopathy and ischemia–reperfusion renal injury [18,19,20]. Treatment with a recombinant human TM (rhTM) containing the three extracellular domains of the protein ameliorates acute kidney injury, hemolytic uremic syndrome, chronic kidney fibrosis with renal failure, pulmonary fibrosis, and allergic bronchial asthma in experimental mouse disease models [14,21,22,23,24]. Administration of rhTM improved renal function and survival in patients with septic disseminated intravascular coagulation and those with acute kidney injury [25]. rhTM was approved in Japan for the treatment of disseminated intravascular coagulation in clinical practice [26]. We have previously reported that treatment with rhTM inhibits transforming growth factor-β1-mediated lung fibrosis and chronic kidney fibrosis with renal failure by inhibiting the apoptosis of parenchymal cells [24,27].

On this basis, here we hypothesized that treatment with rhTM would protect pancreatic islet β-cells from apoptosis and ameliorate glucose intolerance in a DM mouse model.

## 2. Materials and Methods

### 2.1. Animals

C57BL/6 8–10 week-old male mice were purchased from Nihon SLC (Hamamatsu, Japan). Mice were bred in the animal laboratory at Mie University in a pathogen-free environment at 25 °C, with a humidity of about 50%, and they were subjected to a light/dark cycle of 12 h each. Food and water were freely available. The Committee on Animal Investigation of Mie University approved the experimental protocols (approval no. 27-4; date: 19 August 2015), and all procedures were carried out following the institutional guidelines.

### 2.2. Experimental Groups

Streptozotocin (STZ) (Sigma, St. Louis, MO, USA) was injected intraperitoneally to develop diabetes. STZ at a dose of 40 mg/kg body weight was administered for five consecutive days, and the control group was administered the same volume of saline (SAL). Human recombinant thrombomodulin α (rhTM) (provided by Asahi Kasei Pharma Corporation, Tokyo, Japan) at a dose of 3 mg/kg body weight was injected intraperitoneally three times a week for eight consecutive weeks (Figure 1A). The first administration was performed approximately 3 h before STZ injection. The same volume of saline was administered in the non-treated group. Mice were divided into four experimental groups: a group that received intraperitoneal saline and were treated with saline (SAL/SAL), a group that received intraperitoneal saline and were treated with intraperitoneal rhTM (SAL/rhTM), a group that received intraperitoneal STZ and were treated with saline (STZ/SAL), and another group that received intraperitoneal STZ and were treated with rhTM (STZ/rhTM). Each group comprised eight mice.

### 2.3. Diabetes Status Evaluation

Non-fasting blood glucose levels were measured weekly after STZ injection. To perform the glucose tolerance test, glucose was injected intraperitoneally at a dose of 1 g/kg after overnight fasting, and the blood glucose levels were measured after 0, 15, 30, 60, and 120 min. An insulin secretion stimulation test was performed by intraperitoneally injecting glucose at a dose of 3 g/kg after 16 h of fasting, and the blood insulin levels were measured 0, 15, and 30 min after glucose injection. Tail venous blood was collected to measure blood glucose or blood insulin levels. The blood glucose level was measured by the glucose oxidase method. Blood insulin concentration was measured using an ALPCO measuring instrument kit (Salem, NH). The concentration of glucagon was measured using the Glucagon ELISA kit Wako (Sandwich method) from Fujifilm Wako Chemicals Corporation (Osaka, Japan).

### 2.4. Histological Study

Mice were euthanized by an overdose of isoflurane followed by exsanguination eleven weeks after STZ injection. The pancreas was resected, dehydrated, paraffin-blocked, and then cut into 3 µm-thick sections. Immunofluorescence staining of insulin and glucagon was performed at Morpho Technology using specific antibodies. Immunostaining of F4/80-positive cells (macrophages) was performed as previously described [28]. The slides were observed under an optical microscope BX53, and microphotographs were taken using a DP73 digital camera with DP controller software (Olympus, Tokyo, Japan). Quantification of insulin- and glucagon-producing cells was performed using the WinROOF image processing software (Mitani Corporation, Tokyo, Japan).

### 2.5. Apoptosis Evaluation in Pancreatic Islets

Apoptosis of pancreatic islets was evaluated using a terminal deoxynucleotidyl transferase dUTP nick-end labeling (TUNEL) staining kit (Chemicon International, Temecula, CA, USA). Microphotographs of TUNEL-stained islets were taken using an Olympus BX53, DP 73 digital camera and DP Controller software (Olympus, Tokyo, Japan).

### 2.6. Cell Culture

Min6 cells, the murine pancreatic β-cell line, were kindly provided by J. Miyazaki (Osaka University, Osaka, Japan) and cultured in 10% heat-inactivated FCS-containing Dulbecco’s modified Eagle’s medium (DMEM; Sigma-Aldrich, St. Louis, MO, USA).

### 2.7. Apoptosis Assay

Min6 cells were seeded on 12-well plates and cultured up to sub-confluence. The cells were washed with DMEM containing 1% bovine serum albumin, pretreated with 200 nM rhTM, and apoptosis was induced by 5 mM STZ. After 24 h of exposure, the cells were stained with fluorescein isothiocyanate-annexin V and propidium iodide (FITC Annexin V Apoptosis Detection Kit with PI, Biolegend, San Diego, CA, USA) and analyzed by flow cytometry (FACScan, BD Biosciences, Oxford, UK).

### 2.8. Akt Activation

Activation of Akt was evaluated by Western blotting using antibodies specific to phosphorylated Akt and total Akt (Cell Signaling Technology, Danvers, MA, USA). The bands’ intensity was quantitated by densitometry using the public domain NIH ImageJ program (wayne@codon.nih.gov; Wayne Rasband, NIH, Research Service Branch).

### 2.9. Analysis of Spleen Cells

After euthanasia, the spleens from mice of each group were removed, and splenocytes were isolated as previously described [29]. Then, the cells were analyzed with a flow cytometer (FACScan, BD Biosciences, Oxford, UK) using the reagents described in Appendix A. The Cell-Quest Pro software (BD Biosciences) was used for data analysis.

### 2.10. Statistical Analysis

The parameter values are expressed as mean ± standard deviation (S.D.) unless otherwise specified. Statistical differences between two groups were tested by unpaired *t*-test and among three or more variables by analysis of variance (ANOVA) and Tukey’s test. For the statistical analysis, we used GraphPad Prism version 9.0 (San Diego, CA, USA). A *p*-value < 0.05 was considered significant.

## 3. Results

### 3.1. Treatment with rhTM Improved Glucose Intolerance

The blood glucose levels were measured every week for nine weeks after STZ intraperitoneal injection. The blood glucose levels were significantly lower in diabetic mice treated with rhTM (STZ/rhTM) compared to mice receiving only saline (STZ/SAL) (Figure 1B). No changes were observed in non-diabetic mice (SAL/SAL and SAL/rhTM). The IPGT test disclosed a significant decrease in the blood glucose levels in diabetic mice treated with rhTM (STZ/rhTM) 60 min after glucose injection compared to mouse counterparts treated with saline (STZ/SAL). The blood glucose levels were also decreased in mice receiving rhTM after 120 min of glucose injection compared to controls, although the decrease did not reach statistical significance. In addition, the area under the curve was also significantly decreased in STZ/rhTM mice compared to STZ/SAL groups. There was no difference between SAL/SAL and SAL/rhTM groups (Figure 1C,D). During the insulin secretion test, insulin levels were significantly enhanced in diabetic mice treated with rhTM before and after 30 min of intraperitoneal glucose injection compared to diabetic mice treated with saline alone (Figure 1E). The concentration of plasma glucagon was also significantly reduced in diabetic mice treated with rhTM compared to untreated mice (Figure 1F). These results suggest the protective activity of rhTM against glucose intolerance.

### 3.2. Increased Insulin-Producing Cells in Diabetic Mice Treated with rhTM

We compared the total area of pancreatic islets between diabetic mice treated with rhTM (STZ/rhTM) or saline (STZ/SAL) and found that rhTM-treated diabetic mice have a significantly larger area of islets than counterpart mice treated with saline (Figure 2A,B). No significant difference was observed between the control (SAL/SAL, SAL/rhTM) groups. We then compared the area of insulin- and glucagon-producing cells in all groups by immunofluorescence staining. The area of β-cells was significantly increased while the area of α-cells was significantly reduced in diabetic mice receiving rhTM compared to diabetic mice treated with saline (Figure 3A–C). There was no difference between the SAL/SAL and SAL/rhTM groups. These findings suggest the protective activity of rhTM on β-cells.

### 3.3. Decreased Islet Infiltration of Macrophages in Diabetic Mice Treated with rhTM

The infiltration of macrophages in pancreatic islets was evaluated by F4/80 immunostaining. The percentage of areas positively stained with anti-F4/80 antibody was significantly higher in untreated diabetic mice (STZ/rhTM) compared to non-diabetic (SAL/SAL) mice. However, the area showing positive staining with F4/80 antibody was significantly reduced in diabetic mice treated with rhTM (STZ/rhTM) compared to untreated mice with diabetes (Figure 4A,B).

### 3.4. rhTM Activated the Akt Pathway and Inhibited Apoptosis of Islet β-Cells

We hypothesized that the beneficial effect of rhTM depends on its antiapoptotic activity. To demonstrate this, we evaluated apoptosis by the TUNEL method. The results showed that diabetic mice treated with saline had significantly increased apoptotic cells compared to diabetic mice treated with rhTM (Figure 5A,B). The area of apoptosis was not different between SAL/SAL and SAL/rhTM groups. We then assessed whether the anti-apoptotic activity of rhTM in vivo was reproducible in vitro using the Min6 β-cell line. We cultured and treated Min6 cells with STZ or SAL in the presence or absence of rhTM 24 h before evaluating apoptosis by flow cytometry. Apoptosis was significantly inhibited in cells pre-treated with rhTM compared to untreated cells (Figure 5C,D). Control cells cultured in the presence of saline and pre-treated with rhTM or SAL showed no changes. Activation of the Akt signaling pathway promotes cell survival [30]. Therefore, we evaluated whether rhTM inhibited the apoptosis of β-cells by increasing *p*-Akt. Min6 cells pre-treated with rhTM or SAL were stimulated with STZ or saline, and after 24 h, *p*-Akt was evaluated by Western blotting. The results showed that treatment with rhTM significantly increased *p*-Akt compared to cells treated with saline (Figure 5E). These findings suggest that rhTM protects β-cells from apoptosis.

### 3.5. rhTM Regulated the Immune Response Under Diabetic Conditions

Previous studies have shown that rhTM modulates excessive immune responses by inhibiting the activation of several immune cells, including dendritic cells, eosinophils, and T cells [13,14,16]. To assess the effect of rhTM on the systemic immune response in our present STZ-induced DM mouse model, we compared the percentage of spleen immune cells between diabetic mice treated with rhTM or SAL. We found a significantly increased percentage of regulatory CD25+/CD4+ T cells and tolerogenic plasmacytoid dendritic cells in diabetic mice treated with rhTM compared to untreated cells (Figure 6). These observations suggest that rhTM modulates the immune response under diabetic conditions.

## 4. Discussion

The present study shows that treatment with rhTM ameliorated glucose intolerance in diabetic mice by protecting insulin-producing cells from apoptosis.

Reduced pancreatic β-cell mass is the ultimate cause of impaired insulin synthesis and secretion in DM [31,32]. Apoptosis of β-cells explains in part the insufficient insulin production and secretion in DM [32,33,34]. Autoimmune-isletitis-associated cell damage in type 1 DM and hyperglycemia-associated oxidative stress and endoplasmic reticulum stress in type 2 DM are involved in pancreatic β-cell apoptosis [35]. A drug that can target and prevent β-cell apoptosis would be an ideal medication for DM. However, antiapoptotic drugs are currently unavailable. In the current study, we evaluated the potential of rhTM as an antiapoptotic drug in DM on the basis of evidence showing its strong anti-apoptotic activity in several organ injury models. Treatment with rhTM inhibited cell apoptosis in experimental animal models of lipopolysaccharide-induced acute kidney injury, hepatic ischemia-reperfusion injury, hepatic sinusoidal obstruction syndrome, cardiopulmonary-bypass-induced acute lung injury, ischemic myocardial injury, atherosclerosis, diabetic nephropathy, glomerulosclerosis, and pulmonary fibrosis [18,21,22,23,24,36,37,38,39,40]. In vitro experiments have shown that rhTM suppresses the apoptosis of endothelial cells, alveolar epithelial cells, hepatocytes, hepatic sinusoidal cells, and podocytes [24,36,38,39,40]. Here, we treated diabetic mice with rhTM and evaluated its effect on β-cell apoptosis and glucose intolerance. Consistent with the anti-apoptotic activity of rhTM observed in other disease models, we found significantly increased areas of the pancreatic islet β-cells and decreased β-cell apoptosis in diabetic mice treated with rhTM compared to untreated mouse counterparts. The inhibition of apoptosis by rhTM correlated with a significant improvement of blood glucose levels, glucose tolerance test, and insulin secretion. In addition, rhTM protected the β-cell line Min6 from apoptosis, and in agreement with previous studies surviving β-cells showed increased activation of the Akt pathway [24]. Overall, these findings support the rationale for targeting β-cell apoptosis and suggest the potential application of rhTM for the treatment of DM.

Islet inflammation or isletitis is a common pathological finding in type 1 and type 2 DM [32,33]. Inflammation in type 1 DM results from an autoimmune response to islet β-cells characterized by a predominant infiltration of CD8+ T-cells and less-abundant CD4+ T cells, B cells, and macrophages [32,35]. In type 2 DM, initial compensatory islet hyperplasia occurs in response to insulin resistance followed by a progressive β-cell dysfunction leading to hyperglycemia, increased oxidative stress, and infiltration of pancreatic islets by macrophages and T-cells [31,33,41,42]. Isletitis is also observed in STZ-induced DM [43,44]. In agreement with previous observations, we found decreased infiltration of macrophages in diabetic mice treated with rhTM compared to their untreated counterpart mice. Therefore, besides inhibiting apoptosis, the beneficial effects of thrombomodulin administration in our DM model may also be attributed to its anti-inflammatory and immunomodulatory activity. Previous studies have shown that rhTM inhibits the activation of eosinophils and mast cells and enhances dendritic cells’ tolerogenic activity as well as the proportion of regulatory T cells in lymphoid tissue [14,16,45]. In agreement with these studies, we found here that diabetic mice treated with rhTM had an increased proportion of regulatory CD4+/CD25+ T cells and tolerogenic plasmacytoid dendritic cells in the spleen compared to control mice. It is worth noting here that the suppressor function of CD4+/CD25+ T cells is defective in diabetes [46]. Overall, these observations suggest that modulation of the inflammatory and immune responses may also explain the beneficial effect of rhTM in our DM model.

The clinical use of rhTM was approved to treat disseminated intravascular coagulation in Japan [26]. The dose used in our diabetic mice (3 mg/kg mouse body weight) was many times higher than that used in clinical practice (0.06 mg/kg) for patients with coagulopathy [26]. However, we observed no bleeding complications, hematuria, or lethal complications in our experimental mouse model, suggesting that rhTM is safe even at higher doses. The delivery system of rhTM in patients is intravenous [26]. However, the intravenous root would be unsuitable for chronic delivery in clinical practice. Similar to insulin, the subcutaneous root would be the ideal delivery system in the clinic. In this regard, a previous study has shown that rhTM has excellent bioavailability (74–108%) by subcutaneous injection and that rhTM administered by the subcutaneous route has a longer half-life than that delivered by intravenous injection (16–28 h vs. 5–7 h) [26]. In addition, the subcutaneous administration of rhTM has been very effective in several experimental disease models, including recurrent miscarriage, chronic kidney vascular injury, radiation-induced intestinal damage, necrotizing enterocolitis, and ventilator-induced lung injury [47,48,49,50,51]. Overall, these previous findings, together with the beneficial effect observed in our model, suggest the potential application of rhTM for the chronic treatment of DM. However, further investigations must be performed to confirm our findings.

## 5. Conclusions

This study shows that rhTM, a drug currently used for the treatment of patients with systemic coagulopathy in Japan, ameliorates glucose intolerance and insulin secretion and enhances the proportion of regulatory T cells and tolerogenic dendritic cells in an experimental mouse model of diabetes, suggesting the potential expansion of its indication in clinical practice.

## Figures and Tables

**Figure 1 cells-10-02237-f001:**
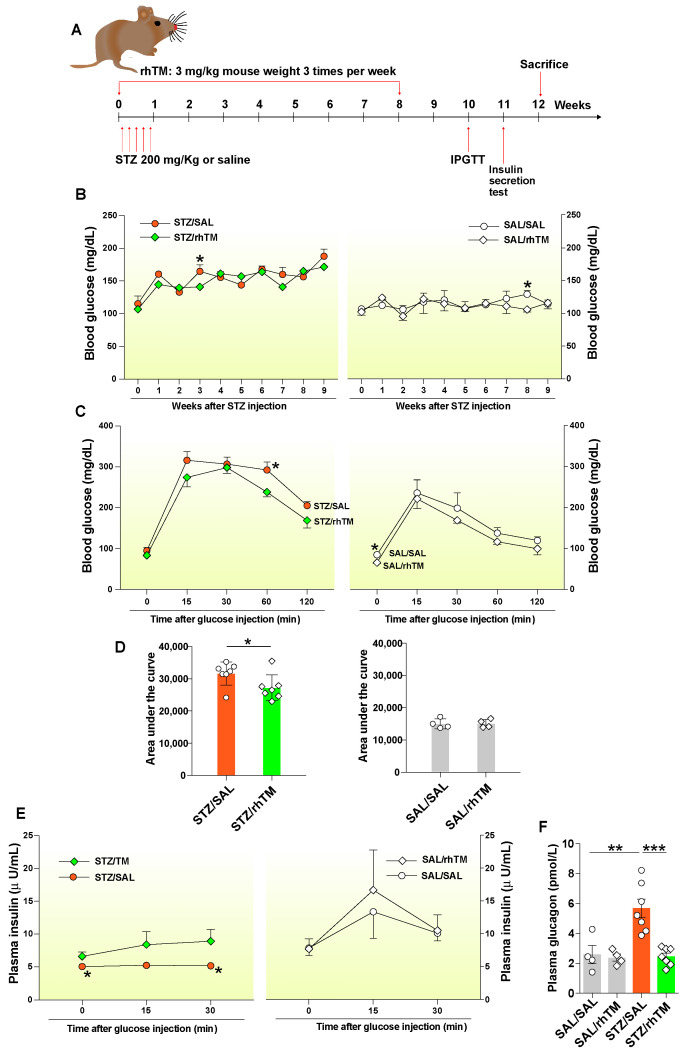
Treatment of diabetic mice with recombinant human thrombomodulin improved glucose tolerance. (**A**) Mice received intraperitoneal injection of streptozotocin (STZ) for five days and were treated with recombinant human thrombomodulin (rhTM) (STZ/rhTM) or saline (STZ/SAL) three times a week for 8 weeks and then sacrificed at 12 weeks. Non-diabetic mice received intraperitoneal saline and were treated with rhTM (SAL/rhTM) or saline (SAL/SAL) for the same time interval and schedule. (**B**) Blood glucose levels were measured as described in the Materials and Methods every week for 9 weeks. (**C**) Intraperitoneal glucose tolerance test was performed on the 10th week after STZ administration, as described in the Materials and Methods. (**D**) The areas under the curve obtained during intraperitoneal glucose tolerance test were calculated. (**E**) The insulin secretion test was performed on the 11th week after STZ administration, as described in the Materials and Methods. (**F**) Plasma glucagon was measured 15 min after glucose challenge as described in the Methods. Data are the mean ± S.D. Statistical analysis by unpaired *t*-test and ANOVA with Tukey’s test. * *p* < 0.05 vs. STZ/rhTM, ** *p* < 0.01, *** *p* < 0.001.

**Figure 2 cells-10-02237-f002:**
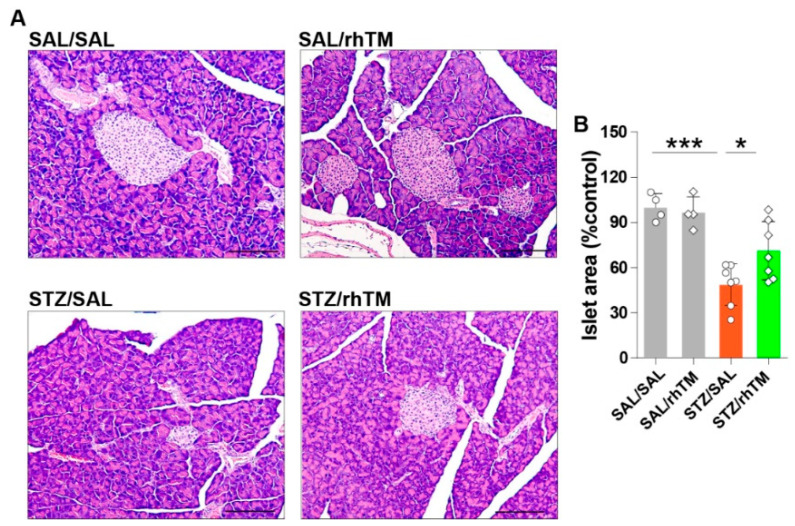
Treatment of diabetic mice with recombinant human thrombomodulin increased the area of pancreatic islets. (**A**) Hematoxylin & eosin staining of the pancreas in each group. (**B**) Area of pancreatic islets measured in each group using the WinROOF image processing software. Scale bars indicate 200 µm. Data are the mean ± S.D. Statistical analysis by ANOVA and Tukey’s test. * *p* < 0.05, *** *p* < 0.001. SAL/SAL, mice received intraperitoneal injection of SAL and were treated with SAL. SAL/rhTM, mice received intraperitoneal injection of SAL and were treated with rhTM. STZ/SAL, mice received intraperitoneal injection of STZ and were treated with SAL. STZ/rhTM, mice received intraperitoneal injection of streptozotocin (STZ) and were treated with recombinant human thrombomodulin (rhTM).

**Figure 3 cells-10-02237-f003:**
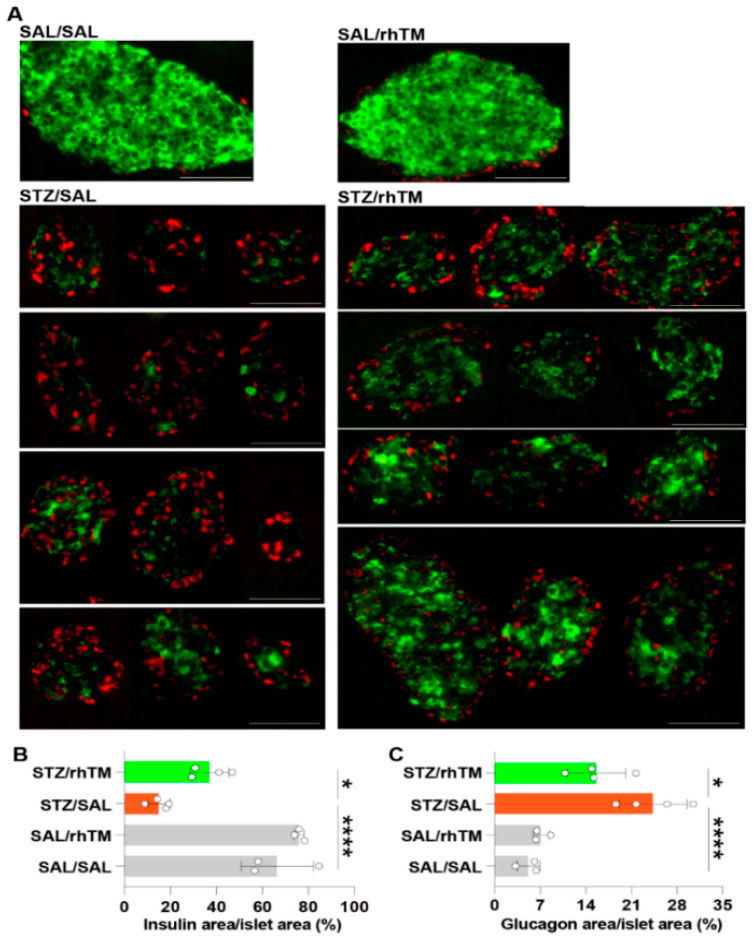
Treatment of diabetic mice with recombinant human thrombomodulin increased the insulin-producing area. (**A**) Immunofluorescence staining of β-cells (green) and α-cells (red) in each mouse group. Scale bars indicate 50 µm. (**B**,**C**) The areas of β-cells and α-cells were measured using the WinROOF image processing software. Data are the mean ± S.D. Statistical analysis by ANOVA and Tukey’s test. * *p* < 0.05, **** *p* < 0.0001. SAL/SAL, mice received intraperitoneal injection of SAL and were treated with SAL. SAL/rhTM, mice received intraperitoneal injection of SAL and were treated with rhTM. STZ/SAL, mice received intraperitoneal injection of STZ and were treated with SAL. STZ/rhTM, mice received intraperitoneal injection of streptozotocin (STZ) and were treated with recombinant human thrombomodulin (rhTM).

**Figure 4 cells-10-02237-f004:**
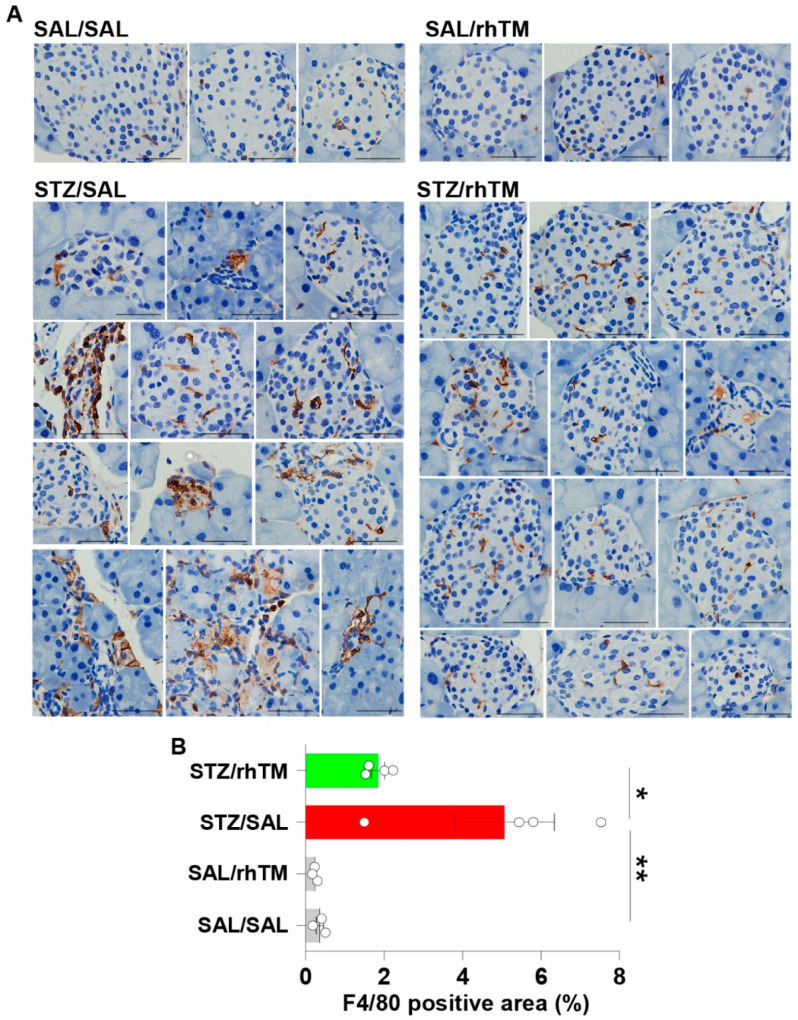
Treatment of diabetic mice with recombinant human thrombomodulin reduced islet infiltration of macrophages. (**A**) Immunostaining of F4/80-positive cells (macrophages) in each mouse group. Scale bars indicate 50 µm. (**B**) The F4/80-positive areas were determined using the WinROOF image processing software. Data are the mean ± S.D. Statistical analysis by ANOVA and Tukey’s test. * *p* < 0.05, ** *p* < 0.01. SAL/SAL, mice received intraperitoneal injection of SAL and were treated with SAL. SAL/rhTM, mice received intraperitoneal injection of SAL and were treated with rhTM. STZ/SAL, mice received intraperitoneal injection of STZ and were treated with SAL. STZ/rhTM, mice received intraperitoneal injection of streptozotocin (STZ) and were treated with recombinant human thrombomodulin (rhTM).

**Figure 5 cells-10-02237-f005:**
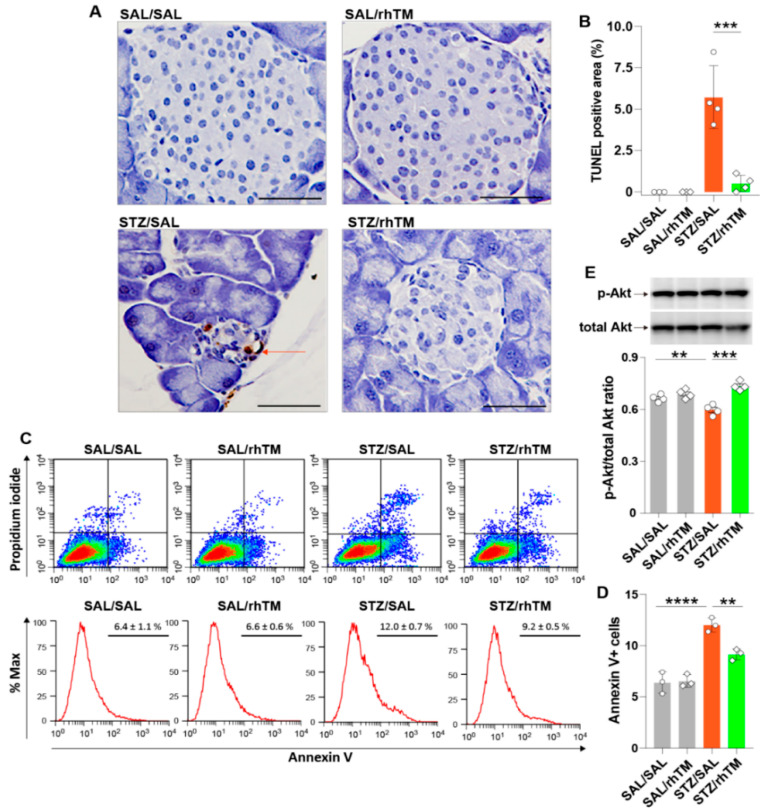
Recombinant human thrombomodulin inhibited apoptosis of pancreatic β-cells in diabetic mice. (**A**) Staining of pancreatic islet cells by TUNEL assay. Scale bars indicate 50 µm. (**B**) TUNEL-positive pancreatic islet areas measured by WinROOF image processing software. SAL/SAL, mice received intraperitoneal injection of saline (SAL) and were treated with SAL. SAL/rhTM, mice received intraperitoneal injection of SAL and were treated with recombinant human thrombomodulin (rhTM). STZ/SAL, mice received intraperitoneal injection of streptozotocin (STZ) and were treated with SAL. STZ/rhTM, mice received intraperitoneal injection of STZ and were treated with rhTM. (**C**,**D**) Min6 cells cultured in the presence of recombinant human thrombomodulin and then treated with STZ for 24 h before evaluating apoptosis by flow cytometry. (**E**) Western blotting of phosphorylated Akt and total Akt in Min6 cells cultured in the presence of rhTM and STZ. Data are the mean ± S.D. Statistical analysis by ANOVA with Tukey’s test. ** *p* < 0.01, *** *p* < 0.001, **** *p* < 0.0001.

**Figure 6 cells-10-02237-f006:**
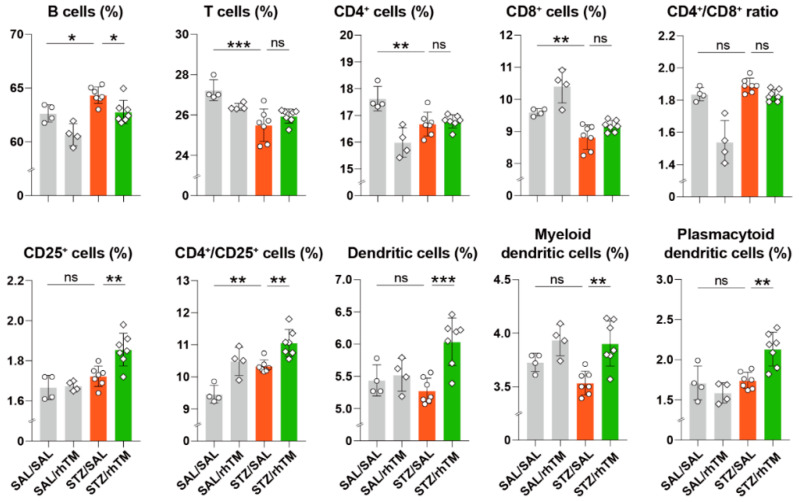
Significant proportion of regulatory T cells and plasmacytoid dendritic cells in spleen from diabetic mice treated with recombinant human thrombomodulin. The spleens from mice of each group were removed, and splenocytes were isolated and analyzed by flow cytometry. Data are the mean ± S.D. Statistical analysis by ANOVA with Tukey’s test. * *p* < 0.05, ** *p* < 0.01, *** *p* < 0.001.

## Data Availability

All data obtained during the current study are available from the corresponding author upon reasonable request.

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
