# Peer review of "Protective Role of Recombinant Human Thrombomodulin in Diabetes Mellitus"

_cells, 2021, doi:10.3390/cells10092237_

Round 1

Reviewer 1 Report

Thrombomodulin, which is predominantly expressed on the endothelium,  has been shown to play an important role in maintaining vascular homeostasis by regulating the coagulation system. Okano et al., show here for the first time that recombinant thrombomodulin (rTM) treatment ameliorates glucose intolerance by protecting beta cells from apoptosis and modulating the immune response in diabetic mice, more a model of type 1 diabetes than type 2 diabetes. This could be better emphasized.  Thus, rTM has the potential to become a novel treatment to improve type 1 diabetes via pleiotropic effects. It would be of interest to see a co-treatment of low-doses of insulin with rTM. The authors should present more images of the islets in Fig. 2C. It is very well established that there is a huge islet heterogeneity, even in mice, and that one can often select the islets (size, etc) they prefer. Plus, this reviewer would like to see the full-length Western blot for Fig. 3E presented in the article. It would also be a plus, to see if rTM treatment reduces hyperglucagonemia after a glucose challenge, which I'm sure was a symptom in these diabetic mice.

Author Response

Response to questions of Reviewer #1

Comment 1:

Thrombomodulin, which is predominantly expressed on the endothelium, has been shown to play an important role in maintaining vascular homeostasis by regulating the coagulation system. Okano et al., show here for the first time that recombinant thrombomodulin (rTM) treatment ameliorates glucose intolerance by protecting beta cells from apoptosis and modulating the immune response in diabetic mice, more a model of type 1 diabetes than type 2 diabetes. This could be better emphasized.  Thus, rTM has the potential to become a novel treatment to improve type 1 diabetes via pleiotropic effects.

Response

We are very thankful to the Reviewer for the constructive and insightful comments that have substantially improved the manuscript.

Comment 2:
It would be of interest to see a co-treatment of low-doses of insulin with rTM.

Response

As the Reviewer suggested, an experiment with co-treatment of low-dose insulin with rTM would be very interesting. Unfortunately, we have not done it in the present study, but we plan to perform it in the following study. 

Comment 3:

The authors should present more images of the islets in Fig. 2C. It is very well established that there is a huge islet heterogeneity, even in mice, and that one can often select the islets (size, etc) they prefer.

Response

As suggested, we have shown more islets in Figure 3 (previous Fig 2C).

Please see Figure 3 on pages 7 and 8, lines 249 to 282 in the revision version of the manuscript.

Comment 4:

Plus, this Reviewer would like to see the full-length Western blot for Fig. 3E presented in the article.

Response

We have uploaded with the manuscript a pdf file containing the full-length Western blot for Fig. 3E.

There appeared to be a mismatch between the blot intensity and the column graphs in the original manuscript. One explanation for this mismatch was that we narrowed too much (from 0.5 to 0.8) the scale of the y-axis of the column graph that made the difference very big. In the revised version of the manuscript, we changed the scale from 0 to 0.9 to show the figure in natural form. We have also chosen more representative blots that better match the graph describing them. We apologize for this.

Please see Figure 5 on page 10, lines 355 to 387 in the revised manuscript.

Comment 5:

It would also be a plus, to see if rTM treatment reduces hyperglucagonemia after a glucose challenge, which I'm sure was a symptom in these diabetic mice.

Response

As suggested, we have measured glucagon and showed the results in Figure 1.

Please see Figure 1F on pages 5 and 6, lines 184 to 221, and the result description on page 6, lines 219 to 220.

Reviewer 2 Report

In this study, authors showed that recombinant human thrombomodulin ameliorates glucose intolerance by protecting pancreatic beta-cells from apoptosis and modulating the immune response in streptozotocin-induced diabetic mice. The concept of this article is interesting and the experiments were well performed. But additional experiments are needed.

Multiple low-dose STZ administration implies that small doses (20 to 40 mg/kg/day) are to be administered over a period of time in order to promote insulitis. During insulitis development, infiltration of macrophages in the pancreatic islet promote cytokine production-dependent T1DM development. Therefore, therapies which target cytokines and nitric oxide are highly probable to be successful in reducing diabetes development in this model.

(References King, A.J. The use of animal models in diabetes research. Br. J. Pharmacol. 2012, 166, 877–894. / Furman, B.L. Streptozotocin-Induced Diabetic Models in Mice and Rats. Curr. Protoc. Pharmacol. 2015, 70, 5.)

Please add the IHC or IF data which show the infiltration of macrophages in the pancreatic islets.

In figure 3E, it seems that the results of the band (phospho-AKT bands in Western blot) and bar graphs do not match. Replace with the representative western blot result most similar to the result of the bar graph.

Author Response

Response to questions of Reviewer #2

Comment 1:

In this paper, Okano Y et al. describe the protecting role of thrombomodulin in insulin-secreting beta cells in a STZ model of T1DM. The experiments are well performed but, several concerns have arisen after reading carefully the manuscript. 

Response

We very much appreciate the constructive comments of the Reviewer that substantially improved our manuscript.

Comment 3:

In figures 1B and 1C, when the animals were treated with thrombomodulin, still there is a high glucose level, Is there a total recovery of glycemia at longer periods of time longer than 12 weeks?

Response

We have not done experiments for more than the 12th week and cannot answer the question. However, rhTM strongly inhibited inflammation and apoptosis in the islets, and therefore, we believe that this beneficial effect would continue for more than 12 weeks and ultimately a decrease in the glucose level would occur.

Comment 3:

Since thrombomodulin has an impact on pancreatic beta cells, Why there are no changes in these cells in control animals, not inducing diabetes?

Response

It is probably because rhTM only exerts antiapoptotic activity under diabetic conditions but not under normal conditions. Thus why we did not find any changes in the normal controls.

Comment 4:

In figure 3E, basal levels of phospho-AKT are very high in the STZ/SAL, where there is a high destruction of beta cells. Any explanation about that?

Response:

As we responded to another reviewer, there appeared to be a mismatch between the blot intensity and the column graphs in the original manuscript. One explanation for this mismatch was that we narrowed too much (from 0.5 to 0.8) the scale of the y-axis of the column graph that made the difference apparently very big. Therefore, in the revised version of the manuscript, we changed the scale from 0 to 0.9 to show the figure in natural form. We have also chosen more representative blots that better match the graph describing them.

The Akt pathway protects against cell apoptosis. Therefore, increased phosphorylation of Akt in the STZ/rhTM indirectly suggests that rhTM is inhibiting apoptosis.

Comment 5:

In figure 4, the analysis was performed in the spleen, did the authors try to analyze directly the islets?

Response:

We used all the pancreatic islets to evaluate for immunostaining, and there was no remaining tissue to evaluate the type of cells, thus why we could not do that experiment.

Reviewer 3 Report

In this paper, Okano Y et al. describe the protecting role of throbomodulin in insulin-secreting beta cells in a STZ model of T1DM. The experiments are well performed but, several concerns have arisen after reading carefully the manuscript. 

Questions:

  1. In figures 1B and 1C, when the animals were treated with thrombomodulin, still there is a high glucose level, Is there a total recovery of glycemia at longer periods of time longer than 12 weeks?
  2. Since thrombomodulin has an impact on pancreatic beta cells, Why there are no changes in these cells in control animals, not inducing diabetes?
  3. In figure 3E, basal levels of phospho-AKT are very high in the STZ/SAL, where there is a high destruction of beta cells. Any explanation about that?
  4. In figure 4, the analysis was performed in the spleen, did the authors try to analyze directly the islets?

Author Response

Response to questions of Reviewer #3

Comment 1:

In this study, authors showed that recombinant human thrombomodulin

ameliorates glucose intolerance by protecting pancreatic beta-cells from

apoptosis and modulating the immune response in streptozotocin-induced

diabetic mice. The concept of this article is interesting and the

experiments were well performed. But additional experiments are needed.

Response

We very much appreciate the constructive comments of the Reviewer that substantially improved our manuscript.

Comment 2:

Multiple low-dose STZ administration implies that small doses (20 to 40 mg/kg/day) are to be administered over a period of time in order to promote insulitis. During insulitis development, infiltration of macrophages in the pancreatic islet promote cytokine production-dependent T1DM development. Therefore, therapies which target cytokines and nitric oxide are highly probable to be successful in reducing diabetes development in this model. (References King, A.J. The use of animal models in diabetes research. Br. J. Pharmacol. 2012, 166, 877–894. / Furman, B.L. Streptozotocin-Induced Diabetic Models in Mice and Rats. Curr. Protoc.Pharmacol. 2015, 70, 5.). Please add the IHC or IF data which show the infiltration of macrophages in the pancreatic islets.

Response:

We have performed the macrophage staining and presented the data in the revised manuscript. We have also added the references suggested by the Reviewer.

 Please see Figure 4 and its legend on pages 8 and 9, lines 283 to 315.

Please see page 3, lines 116 to 117, pages 6 and 7, lines 242 to 248, and page 12, lines 431 to 432 in the revised manuscript.

Comment 3:

In figure 3E, it seems that the results of the band (phospho-AKT bands in Western blot) and bar graphs do not match. Replace with the representative western blot result most similar to the result of the bar graph.

Response:

As we responded to another reviewer, there appeared to be a mismatch between the blot intensity and the column graphs in the original manuscript. One explanation for this mismatch was that we narrowed too much (from 0.5 to 0.8) the scale of the y-axis of the column graph that made the difference apparently very big. Therefore, in the revised version of the manuscript, we changed the scale from 0 to 0.9 to show the figure in natural form. We have also chosen more representative blots that better match the graph describing them.

Please see Figure 5 (Fig 3 in the previous version) in the revised manuscript.

Round 2

Reviewer 2 Report

In this study, authors showed that recombinant human thrombomodulin ameliorates glucose intolerance by protecting pancreatic beta-cells from apoptosis and modulating the immune response in streptozotocin-induced diabetic mice. The concept of this article is interesting and the experiments including revised data were well performed.

Reviewer 3 Report

The authors have answered all the questions arisen by this referee